# Npr3 regulates neural crest and cranial placode progenitors formation through its dual function as clearance and signaling receptor

**Arun Devotta**[†‡], **Hugo Juraver-Geslin**[†§], **Casey Griffin**, **Jean-Pierre Saint-Jeannet***

Department of Molecular Pathobiology, New York University, College of Dentistry, New York, United States

*For correspondence:
jsj4@nyu.edu

[†]These authors contributed equally to this work

Present address: [‡]Regeneron Pharmaceuticals, Tarrytown, United States; [§]Fondation pour la Recherche Médicale, 54 rue de Varenne, Paris, France

**Abstract** Natriuretic peptide signaling has been implicated in a broad range of physiological processes, regulating blood volume and pressure, ventricular hypertrophy, fat metabolism, and long bone growth. Here, we describe a completely novel role for natriuretic peptide signaling in the control of neural crest (NC) and cranial placode (CP) progenitors formation. Among the components of this signaling pathway, we show that natriuretic peptide receptor 3 (Npr3) plays a pivotal role by differentially regulating two developmental programs through its dual function as clearance and signaling receptor. Using a combination of MO-based knockdowns, pharmacological inhibitors and rescue assays we demonstrate that Npr3 cooperate with guanylate cyclase natriuretic peptide receptor 1 (Npr1) and natriuretic peptides (Nppa/Nppc) to regulate NC and CP formation, pointing at a broad requirement of this signaling pathway in early embryogenesis. We propose that Npr3 acts as a clearance receptor to regulate local concentrations of natriuretic peptides for optimal cGMP production through Npr1 activation, and as a signaling receptor to control cAMP levels through inhibition of adenylyl cyclase. The intracellular modulation of these second messengers therefore participates in the segregation of NC and CP cell populations.

## Editor's evaluation

This fundamental work reports the unique finding that specific ligands and receptors in the natriuretic peptide signaling pathway act during early embryogenesis to discriminate between neural crest and cranial placode fates using two distinct mechanisms. This work will be of broad interest to both developmental and cell biologists.

## Introduction

The vertebrate head is characterized by a complex craniofacial skeleton and paired sensory organs. These structures are derived from two embryonic cell populations the neural crest (NC) and cranial placodes (CP), respectively. The NC contributes to the craniofacial skeleton and a subset of cranial ganglia (*Minoux and Rijli, 2010*; *Cordero et al., 2011*), while CP form the anterior pituitary, optic lens, inner ear, olfactory epithelium and several cranial ganglia (*Schlosser, 2010*; *Grocott et al., 2012*; *Saint-Jeannet and Moody, 2014*). Defects in NC and CP development can cause a wide range of human congenital malformations ranging from craniofacial disorders, blindness, deafness and anosmia, to hormone imbalance and orofacial sensory deficits.

NC and CP progenitors are specified at the neural border zone (NBZ) at the end of gastrulation. There is abundant literature indicating that the activity of molecules of the fibroblast growth factor

(Fgf), Wnt and bone morphogenetic protein (Bmp) families must be precisely modulated to specify both NC and CP progenitors in the embryonic ectoderm (*Stuhlmiller and García-Castro, 2012*; *Bae and Saint-Jeannet, 2014*; *Saint-Jeannet and Moody, 2014*; *Schlosser, 2010*). These growth factors in turn activate a unique repertoire of transcription factors that are responsible for initiating a differentiation program unique to these two cell populations (*Simões-Costa and Bronner, 2015*; *Grocott et al., 2012*). Among these transcription factors, Pax3 and Zic1 are especially critical to promote NC and CP fates (*Monsoro-Burq et al., 2005*; *Sato et al., 2005*; *Hong and Saint-Jeannet, 2007*; *Cornish et al., 2009*; *Garnett et al., 2012*; *Milet et al., 2013*; *Jaurena et al., 2015*; *Dubey et al., 2021*). To characterize the molecular events downstream of Pax3 and Zic1, we performed a screen using *Xenopus* animal cap explants expressing varying combinations of these factors and identified several novel regulators of NC and CP fates including natriuretic peptide receptor 3 (Npr3), a component of natriuretic peptide signaling pathway (*Bae et al., 2014*).

Natriuretic peptides constitute a family of three structurally related but genetically distinct paracrine factors, A-type, B-type, and C-type natriuretic peptides, also known as Nppa, Nppb, and Nppc. They mediate their activity by binding three single-pass transmembrane receptors Npr1, Npr2, and Npr3. Nppa and Nppb have a higher affinity for Npr1, while Nppc binds more specifically to Npr2. All three ligands have a similar affinity for Npr3 (*Suga et al., 1992*). A fourth ligand, Osteocrin/musclin (Ostn) was originally identified in a screen for secreted signaling peptides in bone (*Thomas et al., 2003*) and skeletal muscle (*Nishizawa et al., 2004*). Ostn binds Npr3 specifically and with high affinity, competing with other ligands for binding (*Kita et al., 2009*; *Moffatt and Thomas, 2009*). Npr1 and Npr2 are guanylyl cyclase receptors that produce cGMP upon ligand-mediated activation, which in turn activates cGMP-dependent protein kinase G (PKG) and inhibits the activity of the cyclic nucleotide-degrading enzyme, phophodiesterase 3 (PDE3). In contrast, Npr3 lacks guanylyl cyclase activity and is primarily acting as a clearance receptor to regulate local concentrations of natriuretic peptides through receptor-mediated internalization and ligand degradation (*Nussenzveig et al., 1990*; *Potter, 2011*). In addition, the cytoplasmic tail of Npr3 possesses a Gi activator sequences that inhibits adenylyl cyclase (*Anand-Srivastava et al., 1996*; *Murthy and Makhlouf, 1999*). Natriuretic peptides have been implicated in a broad range of physiological processes, regulating blood volume and pressure, ventricular hypertrophy, fat metabolism, and long bone growth (*Potter et al., 2006*), however very little is known on its role during early embryonic development.

Here, we provide evidence that natriuretic peptide signaling plays an important role in embryogenesis controlling NC and CP progenitors formation in the ectoderm. We propose that through its dual function as a clearance and signaling receptor, Npr3 cooperates with other components of the pathway to differentially activate two developmental programs at the neural plate border.

## Results

### Npr3 is a target of Pax3 and Zic1 at the neural plate border

The transcription factors Pax3 and Zic1 are broadly expressed at the NBZ and become progressively restricted to different regions of the ectoderm. *Pax3* is expressed in the presumptive hatching gland, and *Zic1* marks the anterior neural plate, while both factors are co-expressed in the NC-forming region (*Hong and Saint-Jeannet, 2007*). Using gain-of-function and knockdown approaches we, and others, have shown that Pax3 and Zic1 are necessary and sufficient to promote hatching gland and CP fates, respectively, while their combined activity is essential to specify the NC in the embryo and in isolated explants (*Monsoro-Burq et al., 2005*; *Sato et al., 2005*; *Hong and Saint-Jeannet, 2007*; *Milet et al., 2013*). To gain insights into the mechanisms by which Pax3 and Zic1 regulate NC and CP fates, we performed a microarray screen to identify genes activated by Pax3 and/or Zic1 in *Xenopus* animal cap explants (*Hong and Saint-Jeannet, 2007*; *Bae et al., 2014*; *Jaurena et al., 2015*). Among the genes activated by Pax3 and Zic1 we found several well-characterized NC (*snai2*, *sox8* and *foxd3*) and CP (*six1*, *eya1* and *sox11*) genes, as well as novel regulators of NC and CP formation, including natriuretic peptide receptor 3, *npr3/nprc,* a component of natriuretic peptide signaling pathway (*Bae et al., 2014*). This gene is of particular interest because it is uniquely restricted to both the NC and CP forming regions, and this signaling pathway has not been previously linked to the development of these cell populations.

By whole-mount in situ hybridization (ISH), *npr3* is first detected at early neurula stage (NF stage 13) in a crescent shape domain adjacent to the anterior neural plate (*Figure 1A*). During neurulation (NF stage 15–22), *npr3* expression segregates into two domains that encompass the prospective NC and CP (*Figure 1A*). The anterior domain of the NC (cranial NC) displayed a stronger signal as compared to the most posterior regions. Additional domains of expression at these stages include the prospective floor plate (*Figure 1A*). Later in development, *npr3* is maintained in several CP derivatives including the future inner ear (otic vesicle), sensory ganglia (epibranchial placodes), lens and olfactory epithelium (NF stage 32; *Figure 1A*). Additional expression domains include the floor plate and the *zona limitans intrathalamica* or ZLI, two important signaling centers in the central nervous system that secrete sonic hedgehog (Shh) to pattern adjacent tissues, and the pineal gland (*Figure 1B*).

To further demonstrate that Npr3 is a target of Pax3 and Zic1, we analyzed the regulation of *npr3* expression in the embryos by Pax3 and Zic1 using well-characterized morpholino (MO) anti-sense oligos (Zic1MO and Pax3MO; *Monsoro-Burq et al., 2005*; *Sato et al., 2005*; *Hong and Saint-Jeannet, 2007*). Unilateral injection of either MO resulted in a severe reduction of *npr3* expression, while injection of a control MO (CoMO) had no impact on its expression (*Figure 1C and D*). Taken together, these results indicate that *npr3* is restricted to the NC and CP forming regions, and is regulated by Pax3 and Zic1.

## Developmental expression of natriuretic peptide signaling pathway components

Natriuretic peptide signaling pathway is composed of three major related peptides (Nppa, Nppb, and Nppc) and three single-pass trans-membrane receptors (Npr1, Npr2, and Npr3). By qRT-PCR, all three receptors are activated during gastrulation, *npr1* and *npr3* at NF stage 10.5 (early gastrula), and *npr2* at NF stage 12.5 (late gastrula), with their expression increasing gradually as development proceeds (*Figure 1E–G*; *Figure 1—figure supplement 1*). Among the ligands, *nppa* and *nppb* appear most strongly activated at neurula stages (NF stage 14–15) while *nppc* is expressed throughout gastrulation, neurulation and beyond at relatively constant levels (*Figure 1H–J*; *Figure 1—figure supplement 1*). These results suggest that most components of the signaling pathway are present and active during early embryogenesis, consistent with a potential role in NC and CP specification.

## Npr3 is required for the expression of NC and CP genes

We next tested whether Npr3 was required for NC and CP formation. We used a translation blocking MO (Npr3MO) to interfere with Npr3 function. The activity of the MO was confirmed by Western blot of embryos injected with *npr3* mRNA alone or in combination with Npr3MO, using a commercially available anti-Npr3 antibody (*Figure 2A–B*). Unilateral injection of NPR3MO in the animal pole region of two-cell stage embryos resulted in decreased expression of both NC (*snai2* and *sox10*) and CP (*six1* and *foxi4.1*) genes at the NPB (*Figure 2C–D*). The loss of expression of these genes was associated with a lateral expansion of the neural plate territory visualized by *sox2* and a lateral shift of the epidermal marker, *keratin* (*Figure 2C–D*). These results point to a broad requirement of Npr3 for NC and CP gene expression in the developing embryo.

To confirm the Npr3 knockdown phenotype by a non-MO based approach we used AP-811, a high affinity selective Npr3 antagonist (*Veale et al., 2000*). In these experiments, the blastocoel fluid of stage 9 embryos was replaced by 100–120 nl of a 1 mM solution of AP-811 (*Figure 2E*), and the embryos were analyzed for the expression of *snai2* and *six1* at neurula stage (NF stage15), and *twist1* at tailbud stage (NF stage 25). We found a significant and reproducible reduction of *snai2*, *twist1*, *and six1* expression in AP-811-treated embryos as compared to untreated controls (*Figure 2F–G*), or embryos injected with the same volume of water (*Figure 2—figure supplement 1*), further demonstrating the importance of Npr3 in NC and CP development.

Since the expression of several NC and CP genes was diminished upon Npr3 knockdown, we examined whether their loss could be due to an increase in the number of apoptotic cells. Embryos injected with Npr3MO or CoMO were allowed to develop to neurula stage (NF stage 15) when apoptosis was assessed in the dorsal ectoderm by TUNEL staining. We observed a significant increase in the number of TUNEL-positive nuclei in Npr3-depleted vs. control sides, while CoMO injection had no impact on the rate of cell death (*Figure 2—figure supplement 2*). We also asked whether the changes in gene expression could also reflect altered cell proliferation/cell cycle progression by examining

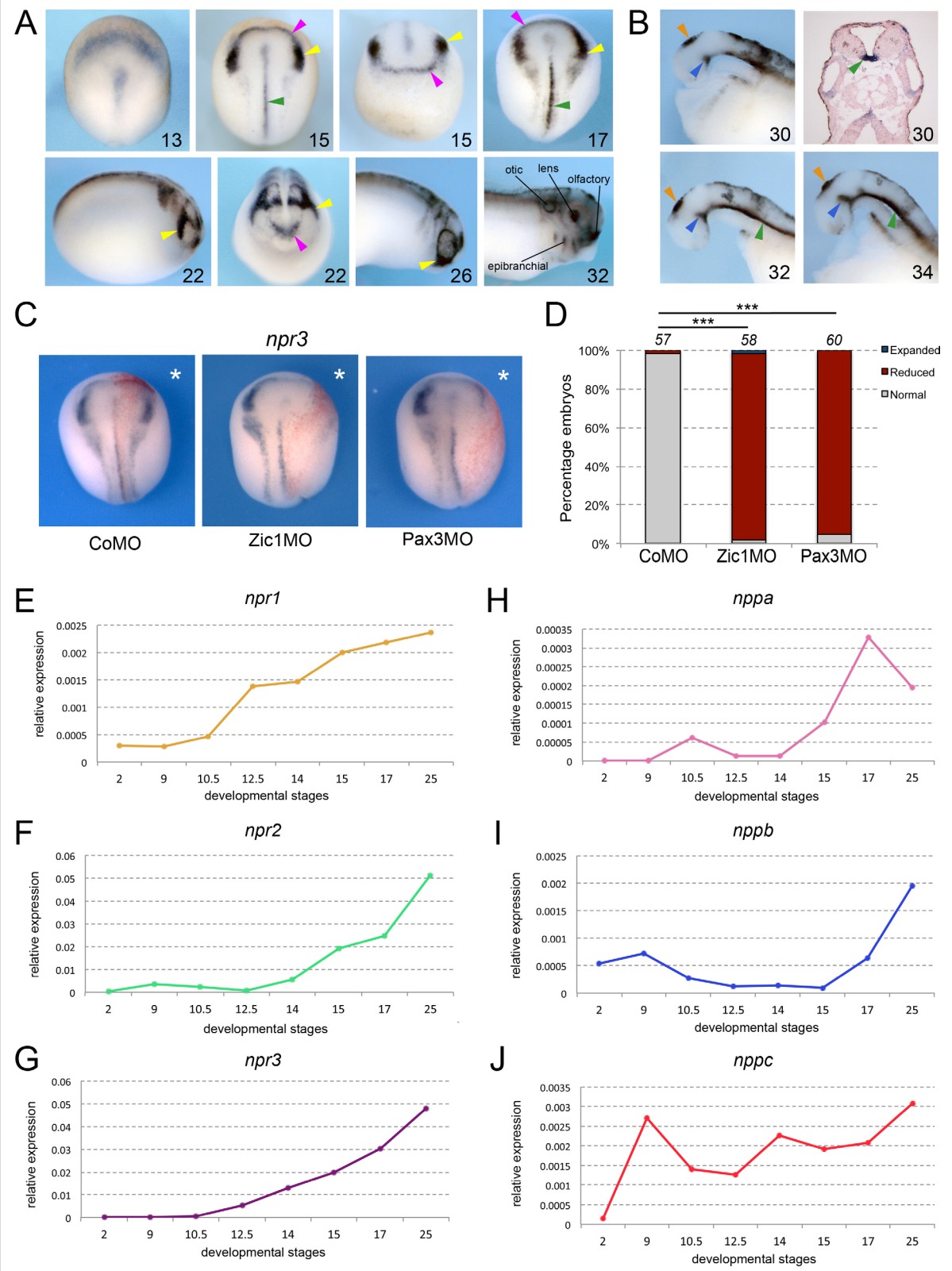

**Figure 1.** Npr3, a target of Pax3 and Zic1, is expressed in developing NC and CP. (**A**) By ISH, *npr3* is detected at the neural plate border at neurula stages (NF stage 13–17), and persists in the NC (yellow arrowheads) and CP (purple arrowheads) forming regions (NF stage 22–26). *npr3* expression is maintained in most CP-derivatives at NF stage 32. (**B**) Dissected brain regions at NF stages 30–34 show additional *npr3* expression domains in the floor plate (green arrowheads), *zona limitans intrathalamica* (blue arrowheads) and pineal gland (orange arrowheads). Upper right shows a transverse section

*Figure 1 continued on next page*

*Figure 1 continued*

through the hindbrain of a NF stage 30 embryo highlighting the floor plate expression of *npr3* (green arrowhead). The embryonic stages are indicated in the lower right corner of each panel. (C) Zic1 (Zic1MO) or Pax3 (Pax3MO) knockdown reduces *npr3* expression at the NPB. Injection of a CoMO did not affect *npr3* expression. The asterisks (*) mark the injected side, also visualized by the lineage tracer Red-gal. Dorsal views, anterior to top. (D) Frequency of the phenotypes. The number of embryos analyzed is indicated at the top of each bar. ***p<0.0005, $\chi^2$ test.(E–J) Developmental qRT-PCR expression profile of *npr1* (E), *npr2* (F), *npr3* (G), *nppa* (H), *nppb* (I) and *nppc* (J) in *Xenopus*. NF developmental stages are indicated on the x-axis, values are normalized to ornithine decarboxylase (*odc*). A representative experiment is shown. Two additional biological replicates are shown in Figure S1.

The online version of this article includes the following figure supplement(s) for figure 1:

**Figure supplement 1.** Developmental RT-PCR of natriuretic peptide signaling pathway components.

the numbers of cells positive for phosphohistone H3 (pHH3). We found a significant decrease in the number of pHH3-positive cells in Npr3-depleted embryos, while CoMO injection did not affect the rate of proliferation (*Figure 2—figure supplement 2*). Altogether these results suggest that Npr3 function is required for survival and cell cycle progression in the ectoderm.

## Clearance vs. signaling activity of Npr3

Npr1 and Npr2 are guanylyl cyclase receptors producing cGMP upon ligand-mediated activation. In contrast, Npr3 lacks guanylyl cyclase activity and is acting as a clearance receptor to regulate local concentrations of natriuretic peptides. In addition, the cytoplasmic tail of Npr3 possesses Gi activator sequence that inhibits adenylyl cyclase (*Figure 3A*; *Anand-Srivastava et al., 1996*; *Murthy and Makhlouf, 1999*). Therefore, Npr3 has both clearance and signaling functions. To distinguish between these activities during NC and CP formation, we performed rescue experiments using mRNA encoding wild-type human NPR3 (HNPR3WT), or a mutated version lacking the cytoplasmic Gi activator sequence (HNPR3ΔC), which abrogates Npr3 signaling function without significantly compromising its clearance activity (*Cohen et al., 1996*). These constructs were first validated by Western blot on lysates of embryos injected with the corresponding mRNAs, and antibodies directed against the N-terminal or the C-terminal region of NPR3 (*Figure 3A and B*). Injection of HNRP3WT mRNA (1 ng) in Npr3-depleted embryos efficiently rescued *sox10* and *six1* expression (*Figure 3C and D*). The same concentration of HNPR3ΔC mRNA (1 ng) was equally efficient at restoring *snai2* expression in morphant embryos (*Figure 3C*), however this truncated molecule was unable to significantly rescue *six1* expression (*Figure 3D*). This result suggests that Npr3 regulates NC and CP fates through different mechanisms.

## Adenylyl cyclase inhibition is critical for the CP-inducing activity of Npr3

Since the expression of HNPR3ΔC in unable to restore *six1* expression in Npr3-depleted embryos, we posited that Npr3 signaling function via inhibition of adenylyl cyclase is critical to mediate the CP-inducing activity of Npr3. To test this possibility, we treated Npr3-depleted embryos with the adenylyl cyclase inhibitor, SQ22536 (50 μM and 100 μM), to determine whether this treatment could restore *six1* expression in Npr3 morphant embryos. We found that in Npr3-depleted embryos the expression of *six1* was significantly rescued upon SQ22536 treatment (*Figure 4A–B*), indicating that the signaling activity of Npr3 is essential for CP formation.

To corroborate this observation, we used an animal cap explant assay and the adenylyl cyclase activator Forskolin (*Figure 4C*). In this assay, Zic1 expression is sufficient to activate *six1* and *eya1* expression (*Figure 4D*; *Hong and Saint-Jeannet, 2007*; *Jaurena et al., 2015*). Upon treatment of these explants with 50 μM Forskolin we found that the induction of *six1* and *eya1* by Zic1 was repressed (*Figure 4D*; *Figure 4—figure supplement 1*). Altogether these observations establish a strong link between Npr3, adenylyl cyclase inhibition and CP gene activation.

## Signaling through Npr1 is required for NC and CP formation

Npr1 is broadly expressed at the early neurula stage, with no apparent tissue restriction, while Npr2 is primarily circumscribed to the prospective trigeminal placode and Rohon-Beard neurons in the trunk (*Figure 5—figure supplement 1*). We next wished to determine whether the two guanylyl cyclase receptors, Npr1 and Npr2, were also implicated in NC and CP formation. We used two

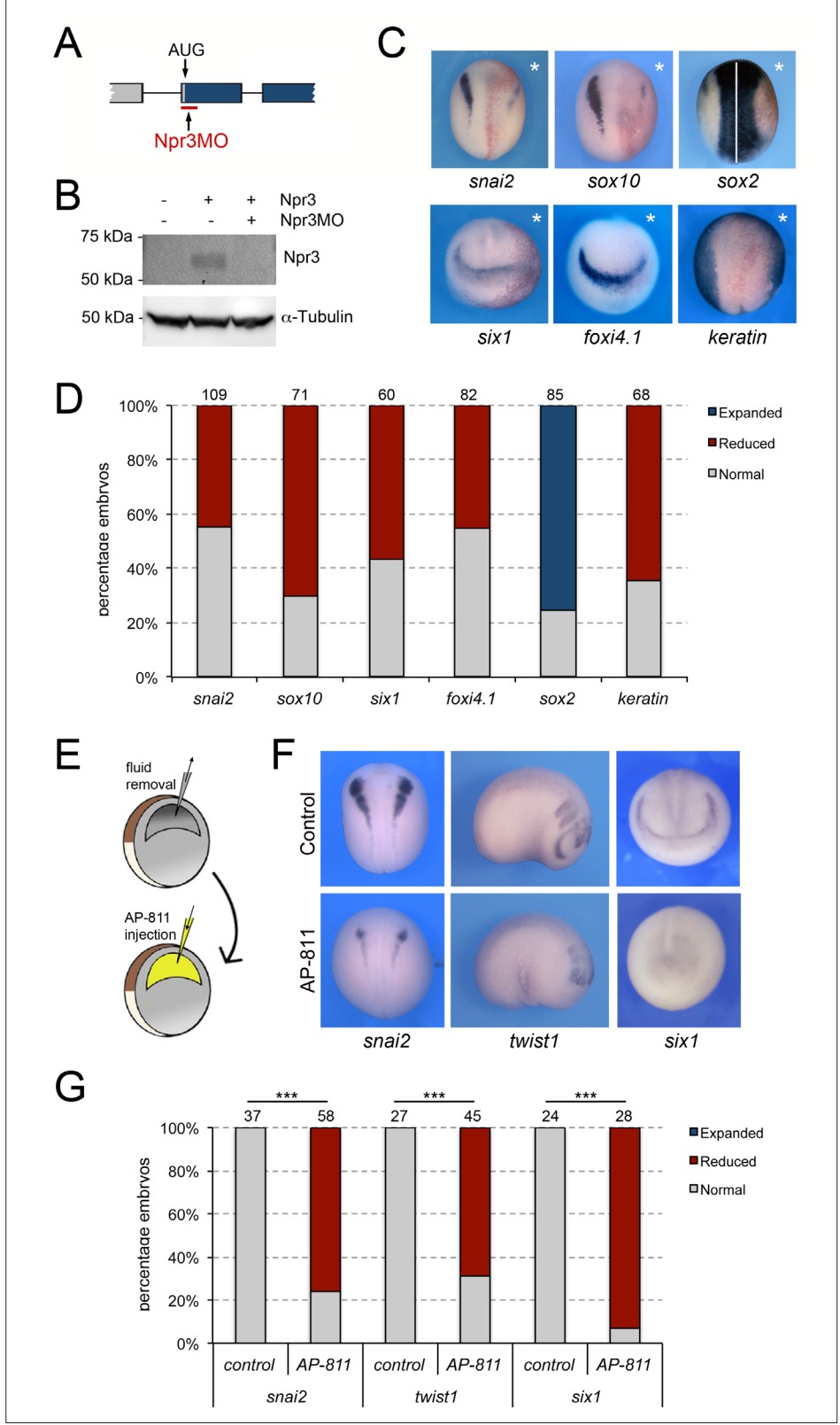

**Figure 2.** Npr3 is required for both NC and CP development. (**A**) Schematic representation of the npr3 gene structure showing the target site of Npr3 translation blocking morpholino (Npr3MO; red). (**B**) Western blot analysis of protein lysates from control embryos and embryos injected with *npr3* mRNA (1 ng) alone or in combination with Npr3MO, using an anti-Npr3 antibody. α-Tubulin is shown as a loading control. (**C**) Phenotype of Npr3MO (30 ng)

*Figure 2 continued on next page*

*Figure 2 continued*

injected embryos. Npr3 knockdown differentially affect the expression of *snai2, sox10, six1, foxi4.1, sox2* and *keratin*. The asterisks (*) mark the injected side, also visualized by the lineage tracer Red-gal. Dorsal views, anterior to top, except for *six1 and foxi4.1*, which show anterior views, dorsal to top. (**D**) Frequency of the phenotypes. The number of embryos analyzed for each condition is indicated at the top of each bar. (**E**) Experimental design for the microinjection of Npr3 antagonist, AP-811, in the blastocoel cavity of blastula stage embryos. (**F**) The expression of *snai2, twist1* and *six1* is reduced in AP-811-treated embryos. Dorsal views, anterior to top (*snai2*). Lateral views, anterior to right (*twist1*). Anterior views, dorsal to top (*six1*). (**G**) Frequency of the phenotypes. The number of embryos analyzed is indicated at the top of each bar. ***p<0.0005, $\chi^2$ test.

The online version of this article includes the following source data and figure supplement(s) for figure 2:

**Figure 2-source data 1**

**Figure 2-source data 2**

**Figure supplement 1.** Manipulation of the blastocoel fluid does not affect *snai2* and *six1 expression*.

**Figure supplement 2.** Npr3-depletion affects the rate of cell death or proliferation in the dorsal ectoderm.

translation blocking MOs to interfere with Npr1 (Npr1MO) and Npr2 (Npr2MO) function, respectively. The activity of each MO was confirmed by Western blot of embryos injected with *npr1 or npr2* mRNA alone or in combination with the corresponding MO, and using commercially available anti-Npr1 and anti-Npr2 antibodies (***Figure 5A***). Unilateral injection of Npr1MO resulted in a marked reduction in *snai2* and *sox10* expression in 72% and 82% of the embryos, respectively (***Figure 5B and C***). Interestingly, *six1* expression was also affected in approximately 46% of injected embryos (***Figure 5B and C***). Similar to Npr3 knockdown, the reduction in NC and CP gene expression was associated with a lateral expansion of *sox2* expression domain (***Figure 5C***). In contrast, Npr2 knockdown (Npr2MO) did not significantly affect the expression of *snai2, sox10* and *six1* (***Figure 5B and C***). These results indicate that Npr2 is not directly involved in the early regulation of NPB cell fates, while Npr1 is required for NC formation and participates in CP formation as well. To ascertain the specificity of the Npr1 phenotype we performed rescue experiments by expression of human NPR1 (HNPR1) in Npr1 morphant embryos. While HNPR1 injection did not significantly affect endogenous *sox10* expression, it restored *sox10* expression in approximately 46% of Npr1-depleted embryos (***Figure 5D–E***).

To confirm these observations by a non-MO based approach we used A71915, a potent and competitive NPR1 antagonist (***Delporte et al., 1992***). Again, in these experiments, the inhibitor was injected in the blastocoel of NF stage 9 embryos (100–120 nl of a 1 mM solution), and the embryos analyzed for the expression of *snai2* and *six1* at neurula stage (NF stage15), and *twist1* at tailbud stage (NF stage 25). We observed a significant reduction of *snai2, twist1 and six1* expression in A71915-treated embryos as compared to controls (***Figure 5F–G***), further demonstrating the importance of Npr1 in NC and CP development.

## Npr1 and Npr3 activation by natriuretic peptides

To evaluate the respective contribution of the three natriuretic peptides to NC and CP formation we used translation blocking MO antisense oligonucleotides directed against each peptide (NppaMO, NppbMO, and NppcMO). While Nppb knockdown did not significantly affect the expression of *snai2, sox10* and *six1*, all three genes showed a marked decrease in expression upon Nppa or Nppc knockdown (***Figure 6A–B***) suggesting that both factors can signal to regulate NC and CP gene expression. Nppa and Nppc morphant embryos also show a broadening of *sox2* expression domain consistent with the reduction in NC and CP gene expression (***Figure 6A–B***). The specificity of the morphant phenotype was confirmed in a rescue assay where injection of recombinant NPPC in the blastocoel of Nppc morphant embryos (***Figure 6C***) was sufficient to restore *six1* expression (***Figure 6D–E***). Spatially, *nppa* is weakly expressed at neurula stage while *nppc* is enriched at the neural plate border (***Figure 5—figure supplement 1***). Later in development, *nppa* expression is confined to the developing heart (***Small and Krieg, 2000***; ***Small and Krieg, 2003***), and *nppc* to the head region including the pharyngeal arches (***Figure 5—figure supplement 1***). Altogether these data suggest that Nppc may represent the primary natriuretic peptide active at these early embryonic stages.

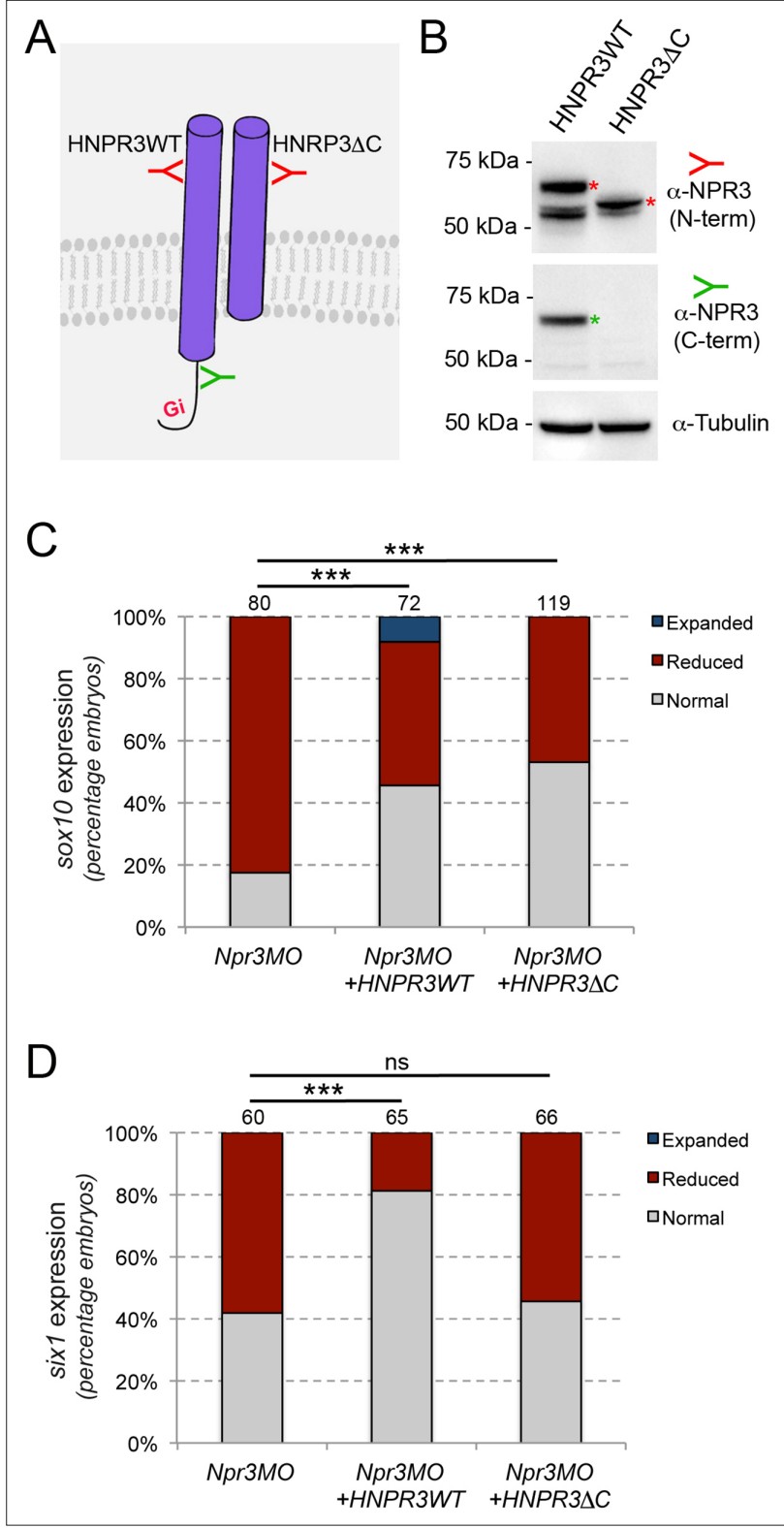

**Figure 3.** Npr3 regulates NC and CP fate through different mechanisms. (**A**) Two expression constructs were generated, a wild-type human NPR3 (HNPR3WT) and a truncated version of HNPR3 lacking the C-terminal Gi activator sequence (HNPR3ΔC). (**B**) Validation of the two constructs. Embryos injected with *HNPR3WT* or *HNRPCΔC* mRNA were analyzed by Western blot using antibodies directed against the C-terminal (ProSci Ψ, Cat#42–909; green) or N-terminal (Santa Cruz Biotechnology, Cat#sc-515449; red) domains of NPR3. α-Tubulin is

*Figure 3 continued on next page*

*Figure 3 continued*

shown as a loading control. The second band in the HNPR3WT lane may represent a cleaved version of HNPR3 lacking the cytoplasmic tail (not recognized by anti-NPR3 C-term), or alternatively a secreted version of NPR3 lacking the transmembrane domain. (**C**) HNPR3WT and truncated HNRP3ΔC are equally efficient at rescuing *sox10* expression in Npr3-depleted embryos. (**D**) In contrast, only HNPR3WT is capable of restoring *six1* expression in Npr3-depleted embryos. (**C,D**) Phenotypes quantification is as described in *Figure 2*. The number of embryos analyzed is indicated at the top of each bar. ***p<0.0005, $\chi^2$ test. ns, not significant.

The online version of this article includes the following source data for figure 3:

**Figure 3-source data 1**

**Figure 3-source data 2**

## Discussion

In the present study, we reveal a novel role for natriuretic peptide signaling in the regulation of NC and CP fates in the ectoderm. Among the components of this signaling pathway, we demonstrate that Npr3 plays a pivotal role by differentially regulating two developmental programs through its dual function as clearance and signaling receptor. Using a combination of MO-based knockdowns, rescue assays and pharmacological inhibitors we show that Npr3, Npr1 and Nppa/Nppc cooperate in the regulation of NC and CP formation, pointing at a broad requirement of this signaling pathway in early embryogenesis. Because of the nature of these receptors, we propose that Npr3 function in the NC territory is to regulate the local concentrations of natriuretic peptides through its clearance activity for optimal cGMP production through Npr1 activation, while Npr3 signaling activity in the CP territory regulates intracellular cAMP levels via inhibition of adenylyl cyclase. Therefore, the proper modulation of these two second-messengers participates in the segregation of tnese two cell populations at the neural plate border. This represents a completely novel function for Npr3 and natriuretic peptide signaling.

Natriuretic peptides have been implicated in a broad range of physiological processes, regulating blood volume and pressure, ventricular hypertrophy, fat metabolism, and long bone growth (*Potter et al., 2006*). In mouse, deletion of *Npr3* causes systemic hypotension and skeletal deformities, including hunched backs, dome-shaped skulls, elongated long bones and vertebral bodies (*Matsukawa et al., 1999*; *Jaubert et al., 1999*). Npr1 null animals show hypertension, ventricular fibrosis and cardiac hypertrophy (*Lopez et al., 1995*; *Oliver et al., 1997*), while Npr2 mutant animals are affected by dwarfism, due to impaired endochondral ossification, and female infertility (*Tamura et al., 2004*). In a recent study, Npr1 has been proposed as a potential therapeutic target for treating acute and chronic itch, as ablation of Npr1-expressing spinal interneurons greatly decreased scratching in responses to itch-inducing agents in mice (*Mishra and Hoon, 2013*; *Solinski et al., 2019*). While there is abundant literature on the functions of natriuretic peptide signaling in these important physiological processes, only a few studies have explored its role in embryogenesis. There is evidence that Npr1 may regulate self-renewal and pluripotency in mouse ESCs (*Abdelalim and Tooyama, 2011*; *Magdeldin et al., 2014*), while Npr3 has been proposed to maintain ESCs function and viability by controlling p53 levels (*Abdelalim and Tooyama, 2012*). Consistent with this finding, we observed increased apoptosis in Npr3-depleted *Xenopus* embryos. During zebrafish heart development, the differential activation of Npr3 or Npr1/Npr2 by varying concentrations of natriuretic peptides can promote or inhibit cardiomyocyte proliferation, respectively (*Becker et al., 2014*). An similar inhibitory activity of Nppa-mediated activation of Npr1 was reported in the context of mouse cardiac progenitors (*Hotchkiss et al., 2015*). Another study has also implicated cardiomyocyte-secreted Osteocrin as an important regulator of membranous and endochondral bone formation in the zebrafish craniofacial skeleton (*Chiba et al., 2017*). Finally, Nppc-dependent activation of Npr2/cGMP signaling pathway regulate somatosensory axons branching, a process of fundamental importance for proper neuronal connectivity in the nervous system (Review in *Dumoulin et al., 2021*).

In zebrafish, Nppa and Nppb as well as Npr1 and Npr2 are functionally redundant during early cardiac development (*Becker et al., 2014*). We did not see the same level of redundancy in *Xenopus*, as both NC and CP formation are exclusively dependent on Npr1 function at these early stages. While typically Npr1 binds Nppa and Nppb at higher affinity than Nppc (*Suga et al., 1992*), our

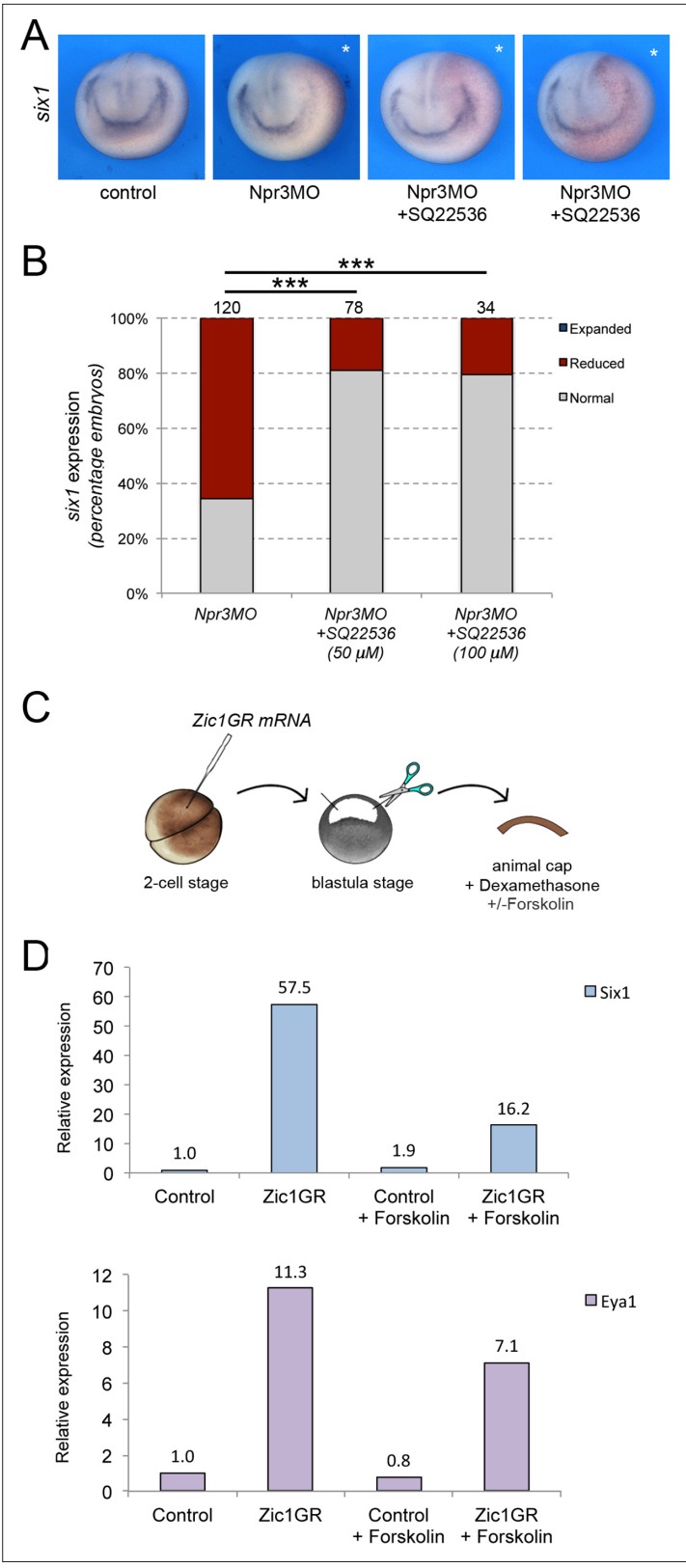

**Figure 4.** Npr3 requires adenylyl cyclase inhibition to promote CP fate. (**A**) Inhibition of adenylyl cyclase by treatment with SQ22536 restores *six1* expression in Npr3-depleted (Npr3MO) embryos. Two representative embryos are shown to illustrate the range of phenotypes. The asterisks (*) mark the injected side, also visualized by the lineage tracer Red-gal. Anterior views, dorsal to top. (**B**) Frequency of the phenotypes. The number of

*Figure 4 continued on next page*

*Figure 4 continued*

embryos analyzed is indicated at the top of each bar. ***p<0.0005, $\chi^2$ test. (**C**) Experimental design to test the effect of adenylyl cyclase activation by Forskolin on the CP-inducing activity of Zic1. (**D**) qRT-PCR analysis of *six1* and eya1 expression in animal cap explants injected with Zic1GR mRNA and cultured in the presence of dexamethasone +/-forskolin (50 μM). Values are normalized to *odc*. A representative experiment is shown. Two additional biological replicates are shown in *Figure 2*.

The online version of this article includes the following figure supplement(s) for figure 4:

**Figure supplement 1.** Forskolin treatment interferes with s*ix1* and *eya1* induction by Zic1GR.

results indicate that both Nppa and Nppc are the active peptides at these early embryonic stages, suggesting some species-specific differences on the relative importance of the players.

The availability and concentration of natriuretic peptides at the cell surface appear to be critical to differentially activate this signaling pathway. This is exemplified by work in zebrafish demonstrating that low and high concentrations of natriuretic peptides can initiate distinct responses in the same cell population. Low concentrations of natriuretic peptides enhance proliferation of embryonic cardiomyocytes in vivo through activation of an Npr3/adenylyl cyclase pathway, while elevated concentrations

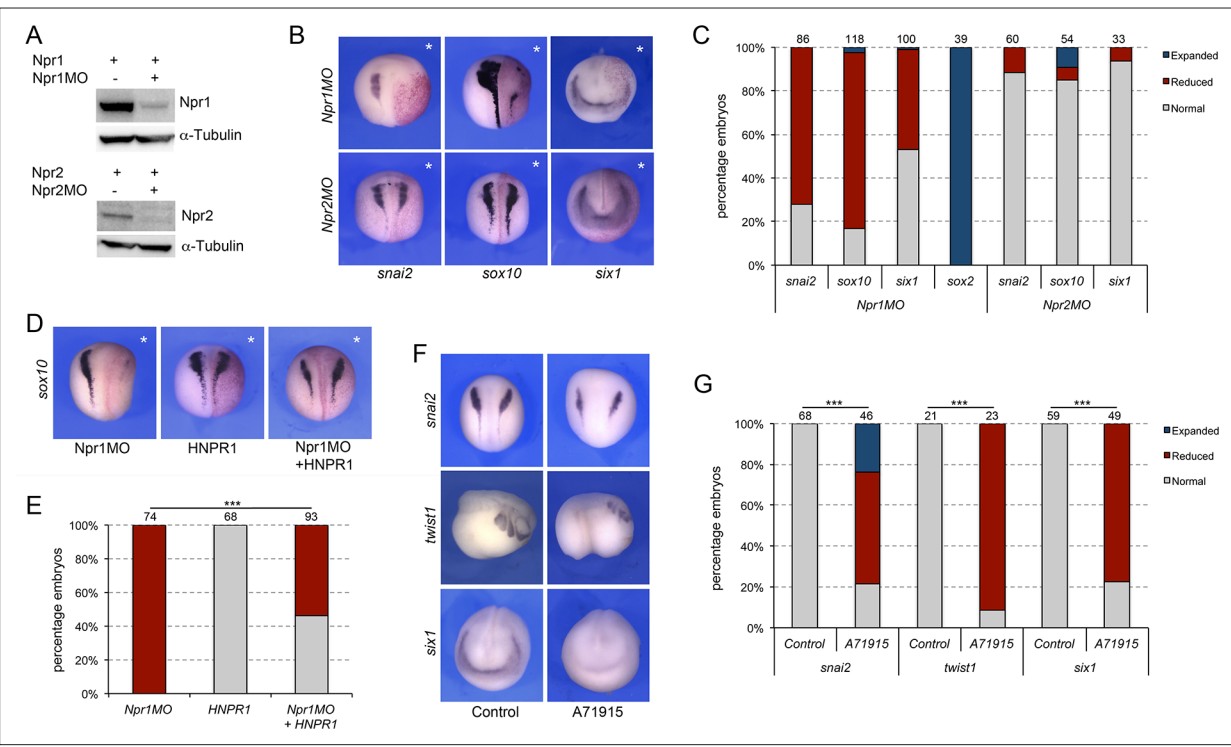

**Figure 5.** Signaling through Npr1 regulates NC and CP formation. (**A**) Western blot analysis of protein lysates from control embryos and embryos injected with *npr1 or npr2* mRNA (1 ng) alone or in combination with the corresponding translation blocking MO, Npr1MO and Npr2MO, respectively. α-Tubulin is shown as a loading control. (**B**) ISH for *snai2, sox10* and *six1* expression in Npr1MO- and Npr2MO-injected embryos. The asterisks (*) mark the injected side, also visualized by the lineage tracer Red-gal. Dorsal views, anterior to top, except for *six1*, which shows anterior views, dorsal to top. (**C**) Frequency of the phenotypes. The number of embryos analyzed for each condition is indicated at the top of each bar. (**D**) Rescue of Npr1 knockdown phenotype (*sox10*) by expression of human NPR1 (HNPR1). The asterisks (*) mark the injected side, also visualized by the lineage tracer Red-gal. (**E**) Frequency of the phenotypes. The number of embryos analyzed is indicated at the top of each bar. ***p<0.0005, $\chi^2$ test. (**F**) The expression of *snai2*, *twist1*, and *six1* is reduced upon treatment with A71915, a potent Npr1 antagonist. Dorsal views, anterior to top (*snai2*). Lateral views, anterior to right (*twist1*). Anterior views, dorsal to top (*six1*). (**G**) Frequency of the phenotypes. The number of embryos analyzed is indicated at the top of each bar. ***p<0.0005, $\chi^2$ test.

The online version of this article includes the following source data and figure supplement(s) for figure 5:

**Figure 5-source data 1**

**Figure 5-source data 2**

**Figure supplement 1.** Spatial expression of *npr1, npr2, nppa* and *nppc* by whole-mount ISH.

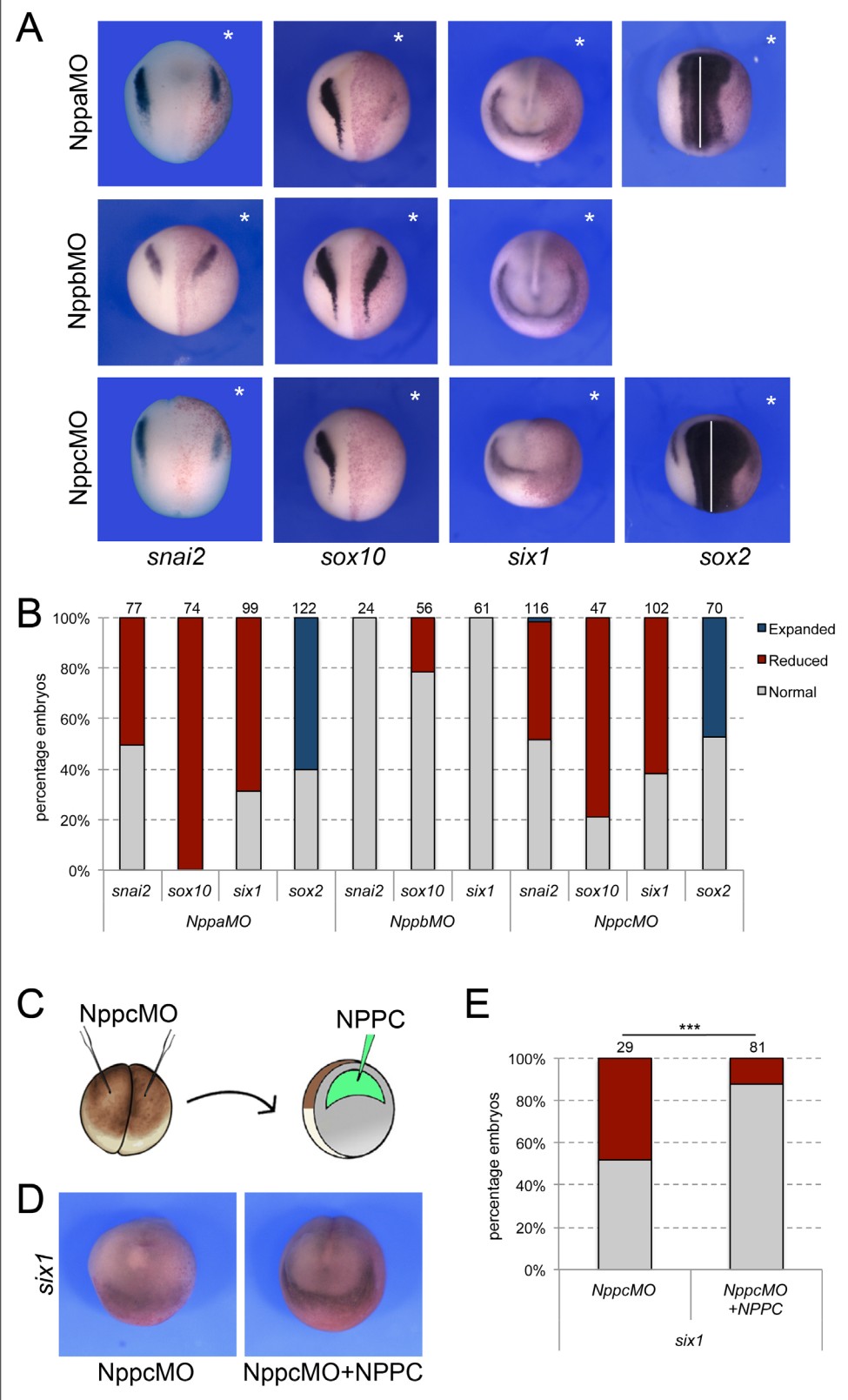

**Figure 6.** Natriuretic peptides activity during NC and CP formation. (**A**) ISH for *snai2, sox10, six1* and *sox2* expression in NppaMO-, NppbMO-, and NppcMO-injected embryos. The asterisks (*) mark the injected side, also visualize by the lineage tracer Red-gal. Dorsal views, anterior to top, except for *six1*, which shows anterior views, dorsal to top. (**B**) Frequency of the phenotypes. The number of embryos analyzed for each condition is

*Figure 6 continued on next page*

*Figure 6 continued*

indicated at the top of each bar. (**C**) To rescue the phenotype of Nppc morphant embryos, recombinant NPPC was microinjected in the blastocoel cavity of Nppc-depleted embryos. (**D**) NPPC can restore *six1* expression in NppcMO-injected embryos. Anterior views, dorsal to top. (**E**) Frequency of the phenotypes. The number of embryos analyzed is indicated at the top of each bar. ***p<0.0005, $\chi^2$ test.

have the opposite outcome, inhibiting their division through activation of Npr1/Npr2 and cGMP production (***Becker et al., 2014***). This observation was further confirmed in mammalian cardiomyocytes cultured in vitro (***Becker et al., 2014***). The role of Npr3 as a modulator of cell proliferation was also described for other cell types with contrasting effects on endothelial and smooth muscle cell proliferation (***Khambata et al., 2011***).

Therefore, local concentrations of natriuretic peptides need to be tightly regulated during development. This can be achieved in various ways, through regulated ligand production, degradation by neutral endopeptidases or via a receptor-mediated clearance mechanism. Our data, using a truncated version of human NPR3 lacking the Gi activator sequences (HNPR3ΔC) to rescue Npr3 knockdown, strongly suggest that the clearance activity of Npr3 is required in the NC lineage to decrease the levels of natriuretic peptides and regulate cGMP production through Npr1 activation. On the other hand, the inability of this truncated construct to rescue CP fate upon Npr3 knockdown is strongly indicative that Npr3 signaling activity and adenylyl cyclase inhibition are necessary for the CP lineage. This was confirmed by restoring CP fate in Npr3-depleted embryos through treatment with SQ22536, a small molecule inhibitor of adenylyl cyclase. Additionally this dual requirement of Npr3 activity sheds light on the importance of cGMP and cAMP, two key second messenger molecules, in establishing NC and CP domains. cGMP through multiple downstream effectors including protein kinases G (PKG), cGMP gated-ion channels and phosphodiesterases can regulate several signaling cascades, while cAMP levels are critical for the activation of cAMP-dependent protein kinase A (PKA) which in turn regulates a broad range of processes such as cell proliferation, apoptosis and differentiation. Future studies will investigate how these second messengers mediate their activity in the specific context of NC and CP development in the embryonic ectoderm.

## Materials and methods
### Constructs and antisense oligonucleotides

*Xenopus laevis* Pax3GR and Zic1GR cloned into pCS2 +are the hormone-inducible versions of Pax3 and Zic1, fused to the human glucocorticoid receptor (GR) ligand-binding domain (***Hong and Saint-Jeannet, 2007***). *Xenopus laevis npr3.L* (Accession number# BC170137) was amplified by PCR with the Pure Taq Ready-to-go PCR beads (ThermoFisher Scientific; Waltham, MA) using NF stage 35 cDNA as template and the following primers (F:TGGAAGGATGTCCTCTATGC; R:ACAG TGCCCCAGTTTTTCAC). The PCR product was subcloned in pGEMT Easy (Promega, Madison WI). This construct is referred as pGEMT-Npr3 (***Bae et al., 2014***). An expression construct containing Npr3 ORF was generated by PCR using pGEMT-Npr3 as template and the following primers (F: GGATCCGCCACCATGTCCTCTATGCT; R:CTCGAGTTATGCTGCAGAAAAATG), and the product was cloned into the expression vector pCS2+. This construct is referred as pCS2 +Npr3. Constructs for *Xenopus laevis nppa.S* (pSPORT6-Nppa; Accession#BC123316), *nppc.S* (pSPORT6-Nppc; Accession#BC076843), *npr1.L* (pSPORT6-Npr1; Accession#BC063739) and *npr2.S* (pCS111-Npr2; Accession #BC129689) were purchased from Dharmacon/Horizon (Lafayette, CO). pSPORT6-Npr1 and pCS111-Npr2 plasmids were used as template to generate by PCR two expression constructs containing the ORF of Npr1 (pCS2 +Npr1; F:CTCGAGGCCACCATGCCTGGAGCTTGGCTT; R: TCTAGATCACCCATGTGTGATGCT) and Npr2 (pCS2 +Npr2; F: ATCGATGCCACCATGGATGT GAAATCTGGC; R: CTCGAGTCAGATAACGCCGTCTTC), respectively. Human NPR3 (HNPR3) construct was a gift from Dr. Pierre Moffat (McGill University; ***Moffatt et al., 2007***). The ORF of HNPR3 was amplified by PCR (F:ATCGATGCCACCATGCCGTCTCTGCTGGTG; R:TCTAGATTAAGC TACTGAAAAATG) from the original construct and cloned into pCS2+ (HNPR3WT). A truncated HNPR3 lacking 34 amino acids at the C-terminal (HNPR3ΔC) was generated by PCR (F:ATCGATGC CACCATGCCGTCTCTGCTGGTG; R: TCTAGATTATTTCTTCCTGAAAAAGTA) and cloned into pCS2+. Human NRP1 (HNPR1) cloned into pCS2 +was purchased from GenScript (Piscataway, NJ).

All expression constructs were confirmed by sequencing and Western blot analysis (see below). In vitro synthesis *of Pax3GR, Zic1GR, GR, Npr1, Npr2, Npr3, HNPR3WT, HNPR3ΔC, HNPR1* and *b-galactosidase* mRNAs from the corresponding pCS2 +expression vector was performed using the Message Machine Kit (Ambion, Austin, TX).

Zic1 (Zic1MO; *Sato et al., 2005*; *Hong and Saint-Jeannet, 2007*), Pax3 (Pax3MO; *Monsoro-Burq et al., 2005*; *Hong and Saint-Jeannet, 2007*), Npr1 (Npr1MO: GGAAAAGCCAAGCTCCAGGCATGA C), Npr2 (Npr2MO: TTTGGTAGCCAGATTTCACATCCAT), Npr3 (Npr3MO: GGCTAAACAAGAGCAT AGAGGACAT), Nppa (NppaMO: AGTATCCAACGAATGAAATCCCCAT), Nppb (NppbMO: GATA GACCTTCCACTCCATTGTAAC) and Nppc (NppcMO: CCCGGTTCCACTTAGTTGCCATCTC) translation blocking morpholino antisense oligonucleotides (MOs) were purchased from GeneTools (Philomath, OR). A standard control MO (CoMO: CCTCTTACCTCAGTTACAATTTATA) was used as control. The efficiency of the translation blocking MOs was evaluated by Western blot analysis (see below).

## Embryos, injections, treatments, and animal cap assay

*Xenopus laevis* embryos were staged according to *Nieuwkoop and Faber, 1967* and raised in 0.1 X Normal Amphibian Medium (NAM; *Slack and Forman, 1980*). The work was performed in accordance with the recommendations of the Guide for the Care and Use of Laboratory Animals of the National Institutes of Health, and was approved by the Institutional Animal Care and Use Committee of New York University. Embryos were injected in one blastomere at the two-cell stage and analyzed by in situ hybridization at stage 15 or stage 25. MO (20–40 ng) were injected together with 0.5 ng of *b-galacto-sidase* mRNA as a lineage tracer. Rescue experiments were performed by sequential injection of the MO and rescuing mRNA resistant to the MO.

For animal cap assay, embryos were injected in the animal pole region in one blastomere at two-cell stage with 250 pg Zic1GR mRNA (*Jaurena et al., 2015*). The explants were dissected at the blastula stage (NF stage 9) and immediately culture in 0.5 X NAM with 10 μM dexamethasone (Sigma-Aldrich, Cat#D1756) for 8 hr. In some experiments, 50 μM Forskolin (Millipore Sigma; Cat#F6886) prepared in DMSO (dimethylsulfoxide; Sigma-Aldrich) was added to the animal cap incubation medium to activate adenylyl cyclase.

## Pharmacological inhibitors and other treatments

For whole embryo treatment with the adenylyl cyclase inhibitor SQ22536 (SantaCruz, Cat#sc-201572), embryos were injected with Npr3MO in 1 blastomere at the two-cell stage, incubated in 50–100 μM of SQ22536 in 0.1 X NAM at NF stage 10.5, and collected at NF stage 15 for ISH analysis. For treatments with the Npr3 antagonist, AP-811 (Tocris, Cat#5498), and Npr1 antagonist, A71915 (Tocris, Cat#6715), approximately 100–120 nl of blastocoel fluid was removed by aspiration at NF stage 9 and replaced with the same volume of water (negative control) or a 1 mM solution of AP-811 or A71915 prepared in water. Embryos were subsequently collected at NF stage 15 and 25 and analyzed by ISH. To rescue the Nppc morphant phenotype, embryos were injected with NppcMO in 2 blastomeres at the two-cell stage. At NF stage 9, approximately 100–120 nl of blastocoel fluid was removed by aspiration and replaced with the same volume of a 1 mM solution of NPPC prepared in water (Sigma-Aldrich, Cat#N8768).

## Whole-mount in situ hybridization

Injected embryos were fixed in MEMFA, stained with Red-Gal (Research Organics; Cleveland, OH) to visualize the lineage tracer (*LacZ*) and processed for in situ hybridization (ISH). Digoxygenin (DIG)-labeled antisense RNA probes (Genius Kit; Roche, Indianapolis, IN) were synthesized using template cDNA encoding *foxi4.1* (*Pohl et al., 2002*), *six1* (*Pandur and Moody, 2000*), *sox2* (*Mizuseki et al., 1998*), *sox10* (*Aoki et al., 2003*), *snai2* (*Mayor et al., 1995*), *keratin* (*Jonas et al., 1985*), *npr3* (*Bae et al., 2014*), *npr1* (pSPORT6-Npr1), *npr2* (pCS111-Npr2), *nppa* (pSPORT6-Nppa) and *nppc* (pSPORT6-Nppc). Whole-mount ISH was performed as previously described (*Harland, 1991*; *Saint-Jeannet, 2017*). Stained embryos were imaged using a dual light-fluorescence Leica M165 Stereomicroscope (Leica Microsystems Inc, North America, Buffalo Grove, IL) and representative images of the dominant phenotype are displayed in the figures.

## Western Blot analysis

To evaluate the efficiency of the MOs, embryos were injected at the two-cell stage with 1 ng of *Xenopus npr1*, *npr2* or *npr3* mRNA alone or in combination with the corresponding MO (20–30 ng) and collected at stage 13. Pools of 10 embryos were homogenized in lysis buffer (1% Triton X-100, 5 mM EDTA in 0.1 x NAM) containing a protease Inhibitors (ThermoFisher Scientific, Cat#78429). After two consecutive centrifugations to eliminate lipids, the lysate was concentrated on an Amicon Ultra Centrifugal Filter (Merck Millipore; Billerica, MA), 10 µl of the concentrated lysate was resolved on a 4–12% NuPAGE Bis-Tris gel and transferred onto a PVDF membrane using the iBlot system (Invitrogen, Waltham MA). Blots were subsequently incubated overnight a 4 °C with rabbit anti-Npr1 (Novus Biologicals, Cat#NBP1-32889, 1:500 dilution), anti-Npr2 (Sigma, Cat#SAB1302067; 1:500 dilution) or anti-Npr3 (Santa Cruz Technology, Cat#sc-130830; dilution 1:100) polyclonal antibodies. The blots were then washed and incubated with goat anti-rabbit IgG-HRP (Santa Cruz Biotechnology, Cat#sc-2004; 1:2,000 dilution) for 1 hr at RT. After several washes, HRP activity was detected with the Western Blotting Luminol Reagent (Santa Cruz Biotechnology, Cat#sc-2048) and imaged on a ChemiDoc MP Biorad gel documentation system (Hercules, CA). Membranes were stripped using Restore Western Blot Stripping Buffer (ThermoFisher Scientific, Cat#21062) and incubated with anti α-tubulin antibody (Sigma Aldrich, Cat#T9026; 1:500 dilution).

To validate HNPR3WT and HNPR3ΔC constructs, two-cell stage embryos were injected with 1 ng of *HNPR3WT* or *HNPR3ΔC* mRNA and collected at NF stage 13. Embryos were extracted and processed for Western blot as described above using antibodies directed against the C-terminal (ProSci Ψ, Cat#42–909; 1 µg/ml) and N-terminal (Santa Cruz Biotechnology, Cat#sc-515449; 1:500 dilution) domains of NPR3, respectively. Each primary antibody was subsequently revealed using donkey anti-goat IgG-HRP (Santa Cruz Biotechnology, Cat#sc2033; 1:5000 dilution) and goat anti-mouse IgG-HRP (Santa Cruz Biotechnology, Cat#sc2005; 1:5000 dilution).

## TUNEL and proliferation assays

TdT-mediated dUTP nick end labelling (TUNEL) staining was conducted as described (*Hensey and Gautier, 1998*). CoMO- and Npr3MO-injected embryos were collected at stage 15 and fixed in MEMFA. End labelling was carried out overnight in TdT buffer containing DIG-dUTP and TdT (Roche, Indianapolis IN). DIG was detected with anti-DIG Fab fragments conjugated to alkaline phosphatase (Roche, Indianapolis IN; 1:2000 dilution). Alkaline phosphatase activity was revealed using NBT/BCIP (Roche, Indianapolis IN) as substrate. For phosphohistone H3 (pHH3) detection (*Saka and Smith, 2001*), fixed embryos were sequentially incubated in anti-phosphohistone H3 antibody (Upstate Biotechnology, Lake Placid NY; 1 mg/ml) and an anti-rabbit IgG secondary antibody conjugated to alkaline phosphatase (ThermoFisher Scientific, Waltham MA; 1:1000) and NBT/BCIP used as substrate. To quantify changes in cell death and proliferation, embryos were individually photographed dorsally and the number of TUNEL- and pHH3-positive cells in the dorsal ectoderm was counted manually by defining two identical rectangular area on each half of the embryos using Adobe Photoshop (*Figure 2—figure supplement 2*), and comparing uninjected and injected sides for both CoMO and Npr3MO.

## qRT-PCR analysis

For the developmental expression analysis, total RNA from 5 embryos was extracted using the RNeasy microRNA isolation kit (Qiagen, Valencia, CA), and the RNA samples were digested on-column with RNase-free DNase I to eliminate genomic DNA, and cDNAs were synthesized using Superscript IV VILO (Invitrogen, Cat#11756050). RT-qPCR was performed using TAQMAN FAST ADVANCED MMIX (Applied Biosystems, Cat#4444557) using custom designed Taqman Probes for Nppa (Assay ID: AP2XD4X), Nppb (Assay ID: APZTJJZ), Nppc (Assay ID: APWC2U2), Npr1 (Assay ID: APPRPKU), Npr2 (Assay ID: AP327PV), Npr3 (Assay ID: AP472AT) and Odc (Assay ID: APCFAEF).

For animal caps, total RNA was extracted from 12 explants using RNeasy microRNA isolation kit (Qiagen, Valencia, CA) and the RNA samples were digested with RNase-free DNase I to eliminate genomic DNA. RT-qPCR analysis was performed using Power SYBR Green RNA to $C_T$ 1 step RT-PCR kit (Applied Biosystems, #4389986) on a QuantStudio 3 Real-Time PCR System (Applied Biosystems, Foster City, CA) using the following primer sets: *Six1* (F: ctggagagccaccagttctc; R: agtggtctcccctca gttt), *Eya1* (F: atgacaccaaatggcacaga; R: gggaaaactggtgtgcttgt), *Sox10* (F:CTGTGAACACAGCATG

CAAA; R:TGGCCAACTGACCATGTAAA), *Snai2* (F: CATGGGAATAAGTGCAACCA; R: AGGCACGT GAAGGGTAGAGA) and *Odc* (F: ACATGGCATTCTCCCTGAAG; R: TGGTCCCAAGGCTAAAGTTG).

## Statistical method

Experiments were performed on embryos obtained from at least three adult females for biological replicates. Only embryos that show overlap between the lineage tracer (*LacZ*) and the ISH signal for the gene of interest were scored. Embryos were scored by comparing the injected side to the uninjected side of each embryo, and to uninjected or CoMO-injected sibling embryos. Embryos were categorized into three phenotypes normal, reduced or expanded based on the overall surface area of the ISH signal for the gene of interest. In all graphs, control groups included both uninjected and CoMO-injected embryos. Significance testing for whole mount ISH experiments was performed using $\chi^2$ test for multiple outcomes. For TUNEL and pHH3 staining p-values were calculated using paired t-test, comparing for each embryo injected and control sides. $P<0.05$ was considered significant. For qRT-PCR experiments, a representative experiment is shown in the main figures, and the two biological replicates are provided as figure supplements.

## Acknowledgements

We are very grateful to Dr. Pierre Moffat (McGill University) and Dr. Mark Hoon (National Institutes Health) for reagents. The work benefited from the support of Xenbase (http://www.xenbase.org/ - RRID:SCR_003280) and the National *Xenopus* Resource (http://mbl.edu/xenopus/ - RRID:SCR_013731). Funded by a grant from the National Institutes of Health to J-P. S-J. (R01-DE025806).

## Additional information

### Competing interests

Arun Devotta: is affiliated with Regeneron Pharmaceuticals. The author has no financial interests to declare. The other authors declare that no competing interests exist.

### Funding

| Funder | Grant reference number | Author |
|---|---|---|
| National Institutes of Health | DE025806 | Jean-Pierre Saint-Jeannet |

The funders had no role in study design, data collection and interpretation, or the decision to submit the work for publication.

### Author contributions

Arun Devotta, Conceptualization, Data curation, Formal analysis, Investigation, Methodology, Writing – original draft, Writing – review and editing; Hugo Juraver-Geslin, Conceptualization, Data curation, Investigation, Methodology, Writing – original draft, Writing – review and editing; Casey Griffin, Data curation, Investigation, Methodology, Writing – review and editing; Jean-Pierre Saint-Jeannet, Conceptualization, Data curation, Formal analysis, Funding acquisition, Investigation, Writing – original draft, Project administration, Writing – review and editing

### Author ORCIDs

Jean-Pierre Saint-Jeannet (iD) http://orcid.org/0000-0003-3259-2103

### Ethics

The work was performed in accordance with the recommendations of the Guide for the Care and Use of Laboratory Animals of the National Institutes of Health, and was approved by the Institutional Animal Care and Use Committee of New York University, protocol #IA16-00052.

### Decision letter and Author response

Decision letter https://doi.org/10.7554/eLife.84036.sa1
Author response https://doi.org/10.7554/eLife.84036.sa2

# Additional files

## Supplementary files
- Transparent reporting form
- Source data 1. Raw data for *Figures 1–6*.

## Data availability
All data generated and analyzed in this study are included in the manuscript and supporting files. Source date files have been provided for all Figures.

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
