## [Editor Report]

This fundamental work reports the unique finding that specific ligands and receptors in the natriuretic peptide signaling pathway act during early embryogenesis to discriminate between neural crest and cranial placode fates using two distinct mechanisms. This work will be of broad interest to both developmental and cell biologists.

---

## [Decision Letter]

**Decision letter after peer review:**

Thank you for submitting your article "Npr3 regulates neural crest and cranial placode progenitors formation through its dual function as clearance and signaling receptor" for consideration by *eLife*. Your article has been reviewed by 3 peer reviewers, and the evaluation has been overseen by a Reviewing Editor and Marianne Bronner as the Senior Editor. The following individuals involved in review of your submission have agreed to reveal their identity: Sally A Moody (Reviewer #1); Dominique Alfandari (Reviewer #3).

Essential revisions:

This manuscript reports the unique finding that specific ligands and receptors in the natriuretic peptide signaling pathway act during early embryogenesis to discriminate between neural crest (NC) and cranial placode (CP) fates. In particular, natriuretic peptide receptor 3 (Npr3) acts as a clearance receptor to regulate local concentrations of natriuretic peptides, and thus optimal cGMP production through Npr1 activation, to promote NC formation. The three reviewers agree that this work is well done and interesting and would be appropriate for publication in *eLife* once some issues have been addressed.

1) The main concern is that the manuscript has not conclusively shown that two different mechanisms are involved in the activation of two distinct developmental programs (i. e. NC and CP progenitor development). The authors interpret their data as showing that the truncated receptor Npr3∆C selectively rescues NC but not CP markers. However it seems possible that the truncated receptor simply has a weaker rescuing activity. It is noted that some embryos injected with the wild-type Npr3 have expanded Six-1 and that is not the case for Npr3∆C (Figure 3C). If Npr3∆C rescuing activity is lower for both NC and CP progenitors, it would contradict the authors' conclusion so experiments addressing this possibility are needed and using an additional marker of CP cells in addition to Six-1 is also important. The authors also presume that this receptor lacks the ability to inhibit cAMP but this should be verified, because Cohen et al., 1996 validated it for a mammalian receptor in a different context.

2)The doses of MOs and human Npr3 mRNA seem unusually high. Can the reason be given?

3) There is also concern about the apoptotosis and cell proliferation assays as they do not seem to be highly specific given the variability observed at the control uninjected side, this should be discussed. Also, neither the text nor the methods mention the area (i.e., length X width) of the embryo examined for the TUNEL and PHH3 staining. The text only says "dorsal ectoderm". When reporting numbers of labeled cells between control and experimental samples, it is important that the same "Area" be examined, for the numbers to be meaningful.

4) In Figures 2F/G, snai2 is used as the NC marker, but in Figure 3C, sox10 is the NC marker. In Figure 5D, sox10 is used and in 5F, snai2 is used. It would be more consistent to use both rather than switching back and forth.

5) In the blastocele injection what was the control? Did the control get 100 nl of water as well? The blascotocell is a complex fluid that contains multiple proteins and ions that are critical for gastrulation.

---

## [Author Response]

Essential revisions:This manuscript reports the unique finding that specific ligands and receptors in the natriuretic peptide signaling pathway act during early embryogenesis to discriminate between neural crest (NC) and cranial placode (CP) fates. In particular, natriuretic peptide receptor 3 (Npr3) acts as a clearance receptor to regulate local concentrations of natriuretic peptides, and thus optimal cGMP production through Npr1 activation, to promote NC formation. The three reviewers agree that this work is well done and interesting and would be appropriate for publication in eLife once some issues have been addressed.1) The main concern is that the manuscript has not conclusively shown that two different mechanisms are involved in the activation of two distinct developmental programs (i. e. NC and CP progenitor development). The authors interpret their data as showing that the truncated receptor Npr3∆C selectively rescues NC but not CP markers. However it seems possible that the truncated receptor simply has a weaker rescuing activity. It is noted that some embryos injected with the wild-type Npr3 have expanded Six-1 and that is not the case for Npr3∆C (Figure 3C). If Npr3∆C rescuing activity is lower for both NC and CP progenitors, it would contradict the authors' conclusion so experiments addressing this possibility are needed and using an additional marker of CP cells in addition to Six-1 is also important. The authors also presume that this receptor lacks the ability to inhibit cAMP but this should be verified, because Cohen et al., 1996 validated it for a mammalian receptor in a different context.

We appreciate the reviewers’ comment regarding the rescuing activity of these constructs. To our defense we would like to point out that (i) the Western blot in Figure 3B indicates that both constructs are expressed at similar levels in the embryo; (ii) the requirement for the signaling activity of Npr3 in CP formation is supported by the rescue assay using an adenylyl cyclase inhibitor, SQ22536; and (iii) this requirement is further corroborated in a different assay using an adenylyl cyclase activator, forskolin, which we show interferes with the expression of 2 CP genes, Six1 and Eya1, in animal caps. Altogether these observations support the conclusions. As far as demonstrating that Npr3∆C lacks the ability to inhibit cAMP, unfortunately we do not have a suitable assay in the embryo for direct validation.

2)The doses of MOs and human Npr3 mRNA seem unusually high. Can the reason be given?

The doses of MOs and mRNAs used in this work are comparable to doses used in our previously published work (Dubey et al., 2021, Timberlake et al., 2021, for example). This is also consistent with other studies that have used similar doses of MOs and mRNAs (Matsuda et al., 2022, Canales Coutino and Mayor, 2022, for example). Furthermore, the ability to rescue the Npr3 KD phenotype together with the confirmation of this phenotype by a non-MO based approach give us confidence in the specificity of Npr3 loss-of-function phenotype at these doses of MOs.

3) There is also concern about the apoptotosis and cell proliferation assays as they do not seem to be highly specific given the variability observed at the control uninjected side, this should be discussed. Also, neither the text nor the methods mention the area (i.e., length X width) of the embryo examined for the TUNEL and PHH3 staining. The text only says "dorsal ectoderm". When reporting numbers of labeled cells between control and experimental samples, it is important that the same "Area" be examined, for the numbers to be meaningful.

Thank you for this comment. We do not have a good explanation for the variability observed at the uninjected side for TUNEL staining, however it is something we have previously reported (see Dubey et al., 2021). In their original paper describing programmed cell death during *Xenopus* development, Hansey and Gautier (1998) report “considerable variation in the degree of TUNEL staining and patterns” between embryos, including left–right asymmetries. They speculate that the presence of a fraction of TUNEL-negative embryos during gastrulation and neurulation may reflect the rapid clearance of dead cells from the embryo. They propose that “embryos showing no staining may be passing through similar waves of cell death, but due to the rapid clearance of dead cells it was not detected.” This phenotype is presumably exacerbated by slight variability in the staging of the embryos. This is now discussed and referenced in the manuscript (see legend for FigS3). Regarding the quantification we are now providing additional information to indicate that the same areas on each side of the embryos (uninjected vs. injected) was used to quantify the number of positive cells (see Materials and methods, page 23, last paragraph), and we now include a visual to illustrate this point (see FigS3A).

4) In Figures 2F/G, snai2 is used as the NC marker, but in Figure 3C, sox10 is the NC marker. In Figure 5D, sox10 is used and in 5F, snai2 is used. It would be more consistent to use both rather than switching back and forth.

To evaluate NC fate in these experiments we have used 3 different markers snai2, sox10 and twist1, and show that they behave in a similar manner in these assays. We appreciate the reviewers’ suggestion, unfortunately as the first authors have now left the lab several months ago, we have not been able to expand the analysis.

5) In the blastocele injection what was the control? Did the control get 100 nl of water as well? The blascotocell is a complex fluid that contains multiple proteins and ions that are critical for gastrulation.

Thank you for pointing that out. We have included control injections in which the blastocoel fluid was replaced with water, the solvent for these inhibitors. In our hands this treatment does not affect gastrulation or the expression of snai2 and six1 (see page 9, first paragraph and FigS2).